# Identification of Black Reef Shipwreck Sites Using AI and Satellite Multispectral Imagery

**Alexandra Karamitrou** [1,*], **Fraser Sturt** [1] **and Petros Bogiatzis** [2]

1    Department of Archaeology, University of Southampton, Southampton SO17 1BJ, UK
2    Ocean and Earth Science, National Oceanography Centre Southampton, University of Southampton, Southampton SO17 1BJ, UK
*    Correspondence: a.karamitrou@soton.ac.uk

**Abstract:** UNESCO estimates that our planet's oceans and lakes are home to more than three million shipwrecks. Of these three million, the locations of only 10% are currently known. Apart from the historical and archaeological interest in finding wrecks, there are other reasons why we need to know their precise locations. While a shipwreck can provide an excellent habitat for marine life, acting as an artificial reef, shipwrecks are also potential sources of pollution, leaking fuel and corroding heavy metals. When a vessel runs aground on an iron-free environment, changes in the chemistry of the surrounding environment can occur, creating a discoloration called black reef. In this work, we examine the use of supervised deep learning methods for the detection of shipwrecks on coral reefs through the presence of this discoloration using satellite images. One of the main challenges is the limited number of known locations of black reefs, and therefore, the limited training dataset. Our results show that even with relatively limited data, the simple eight-layer, fully convolutional network has been trained efficiently using minimal computational resources and has identified and classified all investigated black reefs and consequently the presence of shipwrecks. Furthermore, it has proven to be a useful tool for monitoring the extent of discoloration and consequently the ecological impact on the reef by using time series imagery.

**Keywords:** shipwreck; artificial intelligence; remote sensing; coral reefs; environment; black reefs

## 1. Introduction

The identification and location of shipwrecks, both historic and contemporary, is a priority for a range of stakeholders; from national agencies responsible for quantifying, protecting, and conserving our shared heritage, through to environmental managers [1–3]. Wrecks are complex structures comprised of and containing a range of materials. From an ecological perspective, even small wrecks can pose a threat to wildlife and habitat [4], with the risks increasing with the size of the wreck, construction methods, cargo and fuel type. The physical act of wrecking can cause immediate damage, but gradual degradation through subsequent breaking up and leaching of chemicals can have more severe consequences. As a wreck breaks up, waves and currents can move different parts of a vessel meters to kilometres apart, enlarging the area over which wreck material may be spread and thus the area of potential environmental impact [5,6].

Wrecks, however, pose a challenge for identification from remotely sensed data due to their heterogenous signature and environment of deposition [7,8]. They are often found in shallow or inter-tidal waters, posing a problem for shipborne remote sensing methods due to operational depth constraints. As a wreck breaks up or becomes embedded and buried within a substrate, its visual coherency is disrupted, complicating the process of identification from air and spaceborne sensors. This is significant in light of increasingly common government-led requirements to map and monitor heritage and environment, as well as the growing quantity of remotely sensed data becoming available, through which identification may be attempted.

One route being used to address the need (statutory mapping and monitoring) and opportunity (increased data often at high spatial and temporal resolutions) in coastal monitoring has been the application of machine learning (ML) to track change and identify targets [9,10]. ML enables processing of data at speed and scale. Over the last decade, a considerable amount of research has been published regarding the use of remote sensing methods and artificial intelligence techniques for direct detection of shipwrecks (e.g., [7,11–15]). The challenge that wrecks pose in terms of visible signature, however, remains, with many of these studies requiring clear surface expression and high-resolution data. To circumvent this need, in this paper, we focus on identifying the impact that wrecks have on their environment rather than looking for the structures themselves. This permits identification even in cases where the wreck is not directly visible.

This approach can be adopted when working in remote coral reefs, where the presence of iron is often limited, ranging from 0.2–1 nM [16]. Wreck debris which have high concentrations of iron disturb the local environment by altering the chemistry of the reef. These changes accelerate the growth of invasive organisms such as turf algae, macroalgae, cyanobacterial mats, corallimorphs and other benthic bacterial communities. The result is a discoloration of their surroundings, creating "black reefs" [17,18]. The term black reef reflects the changing of colour of a previously healthy area of reef into different shades between brown and black, due to iron contamination. The development of black reefs has been associated almost exclusively with the presence of wrecks as their development requires relatively large amounts of iron [18].

The exact mechanisms for the development and ecological impact of black reefs have been the subject of increasing interest and research over the last twenty years. This research has intensified due to increased availability of satellite imagery, making the study of remote places feasible. This has led to a growing literature related to the identification, exploration, and ecological consequences of black reefs, mainly in the Pacific Ocean but also in other remote areas [16–27]. Done [28], used the term "phase shift" to describe the change in reef biota from coral to macroalgae, a process that indicates the existence of environmental degradation. In black reefs, during this "phase shift", the coral cover declines to low levels and is replaced by turf algae, macroalgae, cyanobacterial mats and corallimorphs, which among other things reduce the clarity of the overlying water column and alter fish population and diversity [18,19].

In this study, we explore the feasibility of an automated method for locating the presence of shipwrecks in coral reefs. Our approach does not rely on directly detecting the shipwreck, which may be submerged in water and/or sand or broken into small pieces. Instead, we focus on detecting the wider contaminated zone of the black reef in the vicinity of a wreck. To achieve this, we utilise semantic image segmentation by means of artificial intelligence and fully convolutional neural networks, trained using a combination of openly available imagery from Google Earth and commercial images from the Vision-1 satellite with sub-meter spatial resolution. The outcome is not only a method for wreck identification, but also automated remote environmental monitoring.

## 2. Background and Related Literature

The aim of this study is automated identification of black reefs (and by implication, wrecks) in satellite images through semantic image segmentation. The objective is to classify each pixel of an input image into predefined semantic labels that correspond to different feature categories. For each application, the grouping of objects that are associated with a certain label can be different. For example, all pixels that correspond to trees, bushes and grass may be labelled as vegetation. Similarly, sea, lakes, and rivers may be labelled as water, while other categories could be buildings, streets, vehicles etc. Image segmentation is an established and significant component in a wide range of applications, including but not limited to medical imaging [29], autonomous vehicles [30], augmented reality [31], and remote sensing [32,33]. In the latter category, there are many examples from previous work looking to

solve a variety of challenges including environmental monitoring [34], the study of forests [35], agriculture [36], archaeology [32] and land use analysis [37] through segmentation.

Different approaches to segmentation have been trialled based on edge detection [38], thresholding [39], watersheds [40], cluster analysis with k-means [41], graph cuts [42], conditional random fields [43], sparse representations [44] and active contours [45]. Over recent years, however, methods based on deep learning models have been proven to achieve significantly better performance [46–49].

*Deep Learning Models for Feature Detection*

Deep learning is an artificial intelligence method that utilizes artificial neural networks inspired by the structure and function of the human brain. Over the last decade deep learning techniques have transformed research within computer vision, leading to widespread adoption [14,50–52]. Convolutional neural networks (CNNs) are part of the broader family of deep learning methods that are extensively used for image recognition tasks [53,54]. Although an in-depth analysis of the CNN functionality is beyond the scope of this paper, we briefly present some fundamentals relevant to this work.

A CNN is a numerical model that learns directly from data. These processes can either be supervised (where the researcher provides labelled data to train the model) or unsupervised (where the system identifies patterns for itself). For this project, we adopted a supervised learning procedure, providing the system with a set of images labelled with the desired categories. CNN is comprised of several connected convolutional layers. The input layer receives images while the final layer delivers the results in the form of a semantically segmented image where each pixel is classified into one of the predefined categories. The layers in between, which are called hidden layers, carry out the processing. More specifically, a series of feature extracting filters (kernels) are applied to the initial image to extract a set of features (feature maps). These features are then passed on to the next level. The first level extracts basic features (horizontal, diagonal edges etc.) while as one moves deeper into the network, more complex features can be identified. Note that the filters, which are the convolutional layer weights, are learnable parameters of the model, i.e., they are initialized randomly in the beginning and are updated through the optimization procedure that is also called the "learning" or "training" stage.

After the application of each convolutional layer, pooling layers are used that decrease the spatial size of the convolved feature maps by reducing patches of the convolved image into single pixels. In this work we used "max pooling" layers, which resize the convolved image by taking the maximum value of each patch. Pooling layers have no learnable parameters and are followed by layers that implement a sigmoid or ReLU preset activation function, responsible for deciding if the information carried on from the previous layer should be transmitted to the next. As the number of layers increases, the network becomes "deeper" and theoretically more complex patterns can be separated and identified. However, for practical reasons the optimal number of layers generally depends on the amount of labelled data available to train the network. Larger numbers of labelled data provide the potential for resolving more learnable parameters, therefore allowing the inclusion of more hidden layers, which in turn can resolve more complex patterns. In the cases where limited amounts of labelled data are available, the number of trainable layers is kept low.

At present, there is no open access source of labelled data for shipwrecks (or indeed many other forms of archaeological site). As such, there was a need in this study to keep the number of layers and connections low enough to ensure that the optimization problem of network training is not under-determined. Therefore, we used a supervised, fully convolutional, neural network based on the architecture of Semantic Segmentation [55] called SimpleNet [32], that is designed to be implemented for low numbers of labelled data. The network does not contain fully connected layers, and hence requires significantly less memory and computational power [56].

## 3. Study Areas

A shortlist of eight sites suitable for this study was drawn from the published literature (Supplementary Materials). These are listed in Table 1, with the names of the reefs, the number of shipwrecks present and their coordinates (WGS84). From these, seven shipwrecks were used for labelling and training the algorithm and twelve for evaluation. The locations of each site are shown in Figure 1. Seven of the reefs are in the Southwest Pacific, and one in the Indian Ocean.

**Table 1.** Table with the number of Shipwrecks/Black Reefs in each reef and their coordinates in WGS84 system.

| Reef | Number of Shipwrecks | Coordinates (WGS84) | Use |
|---|---|---|---|
| Kanton | 1 | −2.814997, −171.7151 | Training |
| Nikumaroro | 1 | −4.6608, −174.545 | Training |
| Kingmans | 1 | 6.4049, −162.351339 | Training |
| Kenn | 11 | −21.257019, 155.780382<br>−21.254144, 155.780249<br>−21.256865, 155.785203<br>−21.253460, 155.79284 | Training |
| | | −21.266294, 155.751474<br>−21.261727, 155.765001<br>−21.260455, 155.763734<br>−21.260200, 155.766308<br>−21.258671, 155.770358<br>−21.256630, 155.768815<br>−21.258416, 155.771783 | Evaluation |
| Rose | 1 | −14.549244, −168.166467 | Training and Evaluation |
| St. Brandon's | 1 | −16.8309, 59.4756 | Evaluation |
| Kwajalein | 1 | 9.331000, 166.8457 | Evaluation |
| Caroline | 2 | −10.0028, −150.221170 | Evaluation |

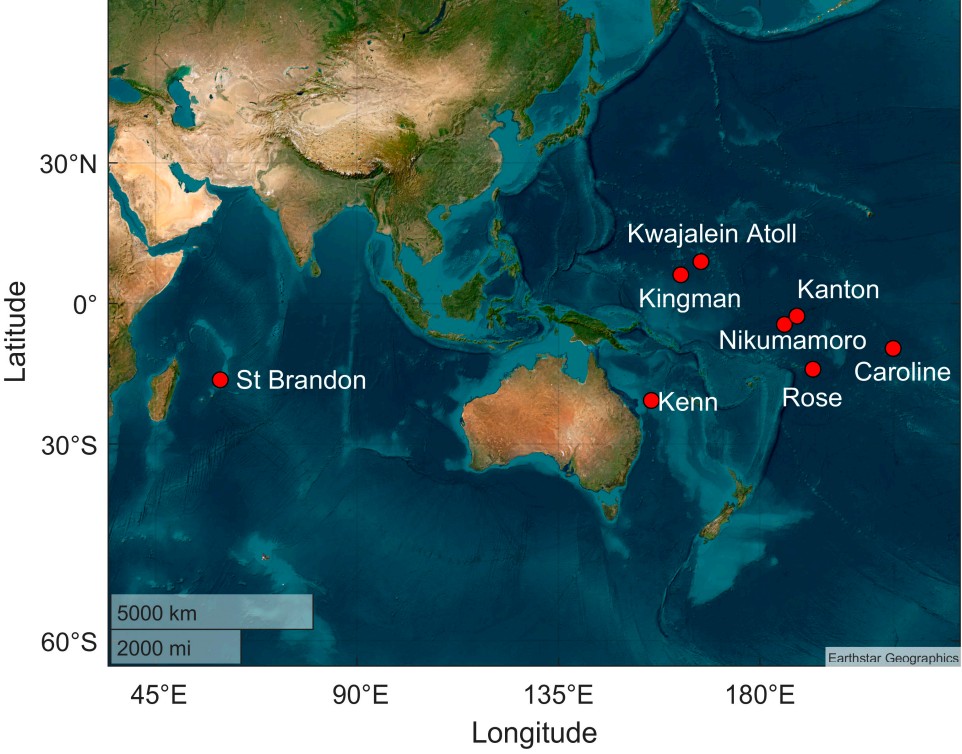

**Figure 1.** Map showing the locations of the reefs investigated in this work.

From this, shortlist sites were then selected for training and evaluation purposes, as not all black reefs are visible in satellite imagery, nor is free imagery available for all reefs. In some cases, the available images were of very low spatial resolution, or had high cloud coverage. Kenn and Rose reefs were selected for both training and evaluation purposes. Kenn reef has 11 known shipwrecks, and thus served as an excellent case study for testing the performance of the approach.

At the Kwajalein reef, we explored a known WWII shipwreck that lies onshore and for which there are currently no reports of the presence of a black reef. A time series of images was also available for this location on Google Earth (from the years 2005, 2013, 2015, 2016, 2019, 2022). This permitted an assessment of the methodology as a tool for environmental monitoring as well as identification.

The following section briefly describes each reef and its corresponding shipwreck/s, with Figure 2 giving imagery for each location.

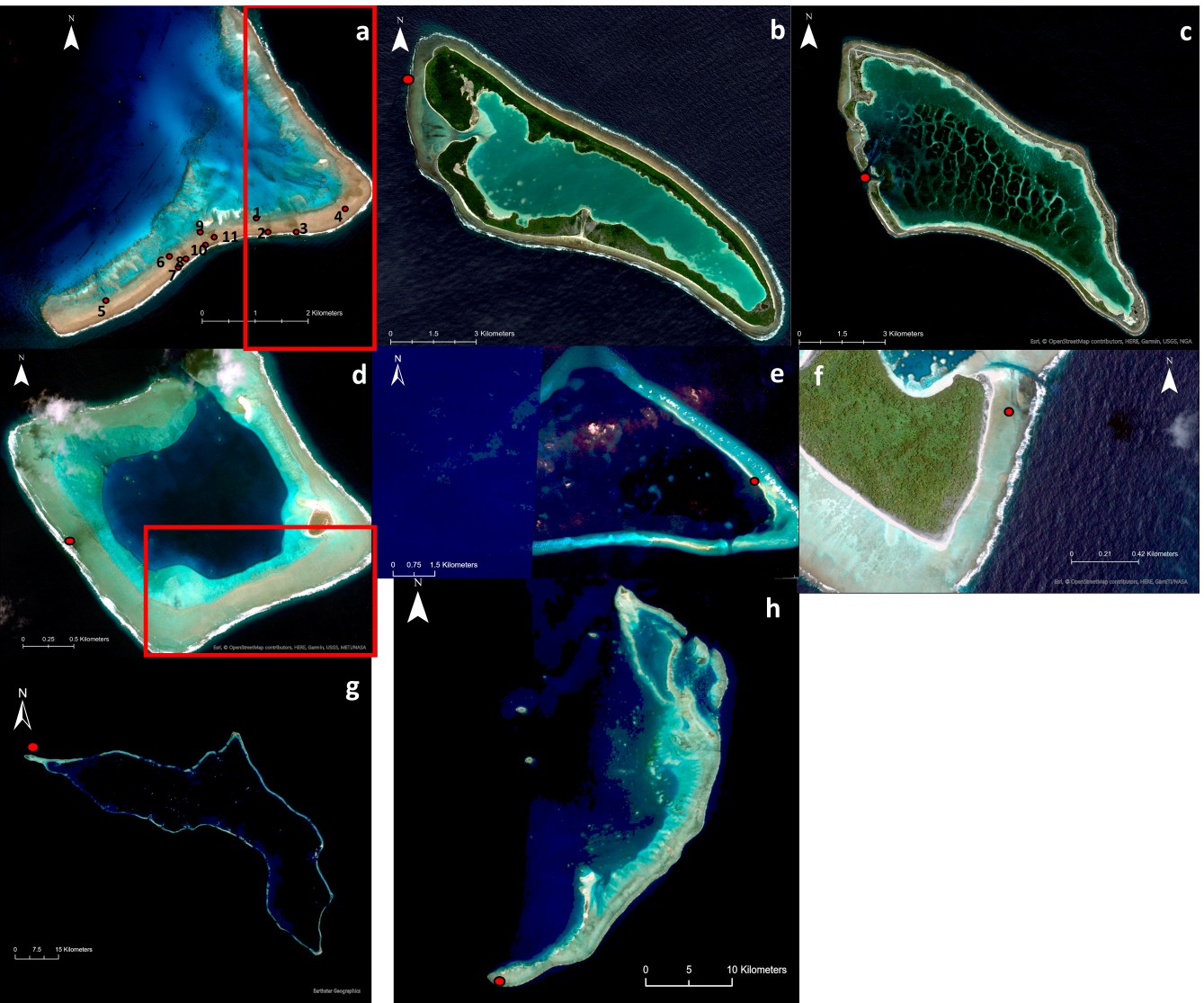

**Figure 2.** Vision-1 imagery from (**a**) Kenn reef, (**b**) Nikumamoro reef and (**c**) Kanton reef. Google Earth imagery from (**d**) Rose reef, (**e**) Kingman reef, (**f**) Caroline reef, (**g**) Kwajalein reef and (**h**) St Brandon reef (Google Earth Pro, viewed 21 May 2021). The red dots show the locations of the shipwrecks in each reef. For the Kenn and Rose reefs, the areas inside the red rectangle represent the part of the image used for labelling and later for evaluation. The rest of the images were used only for evaluation purposes.

### 3.1. Site Descriptions

#### 3.1.1. Kenn Reef

Kenn reef is part of the Coral Sea Islands in the Australian Commonwealth Territory. It is part of a submerged continental block (Kenn Plateau) covering an area of 40 km$^2$ (Figure 2a). The reef has 11 known sites of shipwrecks, 8 of which are 19th century vessels [20]. Discoloration due to the wrecks has developed in all 11 wreck sites of the reef (Figure 2a). Currently, none of these vessels are visible and the only confirmation of their existence is the presence of the discoloration developed in their surrounding environment [57].

One of the known wrecked vessels is the Jenny Lind. She was a single-decked vessel, made from steel, oak and pine with an overall length of 40 m, an 8 m beam and a carrying capacity of 484 tons. Unfortunately, no other information is available regarding the names or type of the vessels that were grounded in the area [20].

#### 3.1.2. Nikumaroro Reef

Nikumaroro reef is part of the Phoenix islands Protected area, in the Republic of Kiribati (Figure 2b).

In 1929, a large, unladen British freighter crashed north-west of the reef [58]. Satellite imagery shows that the body of the vessel has been broken into several pieces that can been seen spread in the surrounding area. As with all the reefs of the Phoenix group, it is a naturally iron-poor region and the artificial introduction of large, iron-rich parts of the wreck has caused discoloration to develop in the surrounding environment.

#### 3.1.3. Kanton Reef

Kanton reef is the largest and northernmost of the eight islands of the Phoenix Islands Protected area, in the Republic of Kiribati, [58] (Figure 2c). It is an iron-poor area where the main source of iron is from marine debris [24]. The appearance of iron-rich materials (mainly from shipwrecks) is linked to the growth of turf algae and benthic bacterial communities, leading to the formation of degraded 'black reefs' [17,18,26].

There are two known shipwrecks which lie stranded on the reef: a whaling ship that was lost in 1854 for which we have no information of its exact location, and the U.S.S. President Taylor that run aground in 1942 [59,60] at the entrance of the west channel; although it is fully submerged, it is still visible from satellite imagery.

#### 3.1.4. Rose Reef

Rose reef is in the south Pacific Ocean and is the principal island of American Samoa (Figure 2d).

On 14 October 1993, a 37 m Taiwanese longline fishing vessel, Jin Shiang Fa, wrecked on the southwest arm of Rose reef. In November the vessel released over 100,000 gallons of diesel fuel and 500 gallons of lube oil. Additionally, more than 300 tons of metallic debris were scattered in the surrounding area, resulting in ecological damage to the reef [61]. In 1999, the Fish and Wildlife Service (FWS) of Honolulu, Hawaii prepared a Restoration Plan leading to the removal of all shipwreck debris by 2010. Although the wreck has been removed, the discoloration that developed due to its presence is still visible through satellite imagery.

#### 3.1.5. Kingman Reef

The Kingman reef is in the North Pacific Ocean and is part of the Line Island chain (Figure 2e). It supports a variety of marine life and is considered to have the second highest coral diversity in the Central Pacific Ocean [62,63].

A teak-hulled fishing vessel was washed into the reef in 2007 and was fully removed in 2014. Initially, the hull of the vessel was located on the fore-reef side of the northeast islet, while in 2010, elements of debris were noticed to have moved in different directions

across the reef [18,64]. The iron materials of the vessel led to the development of visible discoloration in their surrounding areas.

### 3.1.6. Caroline Reef

Caroline reef or Millennium reef is part of the Line Islands in the central Pacific Ocean Republic of Kiribati (Figure 1).

In 1993, a 25 m-long steel tug grounded while trying to tow a sailing vessel out of a narrow reef passage [18]. Although some pictures are provided in the literature [18,65] the exact location of the shipwreck is not clear. However, we can restrict the location of the shipwreck to the southeast entrance of the lagoon (Figure 2f).

### 3.1.7. Kwajalein Reef

Kwajalein reef (Kwajalong) lies in the west-central Pacific Ocean and is part of the Republic of the Marshall Islands (Figure 1). The total land area covers about 16 km$^2$ and forms the world's largest lagoon [66], (Figure 2g).

The atoll was used as a base for the Japanese Imperial Navy during the Second World War, and therefore, many battles took place in close proximity. Today, around twelve shipwrecks and several downed airplanes lie in the surrounding area [67].

At the western part of the atoll, called Ebadon, there is a visible shipwreck that lies on shore at the northmost part of the island.

### 3.1.8. St Brandon Reef

St Brandon reef is part of the Cargados Carajos Shoals and is in the Indian Ocean (Figure 2h). There are no permanent residents on the reef, with only occasional visitors making landfall. The major driver of pollution, therefore, comes from marine debris [27,68,69].

Although various vessels have been stranded on this reef [70], including one which created a concentrated algal bloom in 2012 [27], satellite image coverage is poor. Only one cloud-free Google Earth image (dating to 2005, Figure 2h) is available, prior to the strandings described in [27,70]. Nonetheless, even this single image shows a single-stranded wreck, albeit with the date of loss unknown.

## 4. Materials and Methods

Next, we describe the steps taken to collect, pre-process and compile images for use in training a deep learning model. A general flowchart of this procedure can be seen in Figure 3.

For training purposes, we used two different types of data; four Vision-1 multispectral images (3 channels—red, green, blue) of 3.5 m resolution that were provided free of charge through Jisc (https://www.jisc.ac.uk, accessed on 6 September 2020) and seven Google Earth images (3 channels—red, green, blue) with resolution varying depending on the type of sensor and the year the imagery was acquired (Table 2). The Vision-1 imagery was available for the Kenn, Kanton and Nikumamoro reefs while Google Earth was available for the Rose, Kingman, Kanton, Caroline, St. Brandon, and Kwajalein reefs. The Google Earth imagery was downloaded by using the function provided in the application at an eye altitude of around 5 km and at the maximum available resolution.

To train the algorithm, 256 × 256 × 3-band image tiles were generated from these five reefs (Kenn, Nikumamoro, Kanton, Rose and Kingman) resulting in ~1600 images (Figure 4). To test the performance of the algorithm, we used a small part of the Kenn and the Rose reefs as a training set (Figure 2, part of the images inside the red rectangles) and the rest were used for testing. Figure 4, on the left, shows the Kenn reef (Vision-1, RGB image) and on the right, the Rose atoll (Google Earth, RGB image) with the locations of their shipwrecks as referred to in Table 1. Only subsets of Kenn and Rose reefs were used for training (Figure 2, areas inside the rectangles). These subsets respectively contained 4 (out of 11) and 0 (out of 1) known wrecks. The remaining areas of Kenn and Rose reefs were set aside for evaluation purposes.

**Table 2.** List of all coral reefs and the imagery used in this study.

| Reef | Type of Imagery | Year Acquired | Resolution |
|---|---|---|---|
| Kanton | Vision-1 | 2020 | 3.5 m |
| | Google Earth-Airbus | 2016 | 1.5 m |
| Nikumaroro | Vision-1 | 2020 | 3.5 m |
| Kingmans | Google Earth-Maxar Technologies | 2007 | 0.6 m GSD |
| | | 2013 | 0.5 m GSD |
| Kenn | Vision-1 | 2020 | 3.5 m |
| Rose | Google Earth-Maxar Technologies | 2011 | 0.5 m GSD |
| St. Brandon's | Google Earth-Maxar Technologies | 2005 | 0.65 m GSD |
| Kwajalein | Google Earth-Maxar Technologies | 2005 | |
| | | 2013 | 0.6 m GSD |
| | | 2015 | |
| | | 2016 | |
| | | 2019 | 0.5 m GSD |
| | | 2022 | |
| Caroline | Google Earth-Maxar Technologies | 2004 | 0.5 m GSD |
| | | 2011 | 0.6 m GSD |

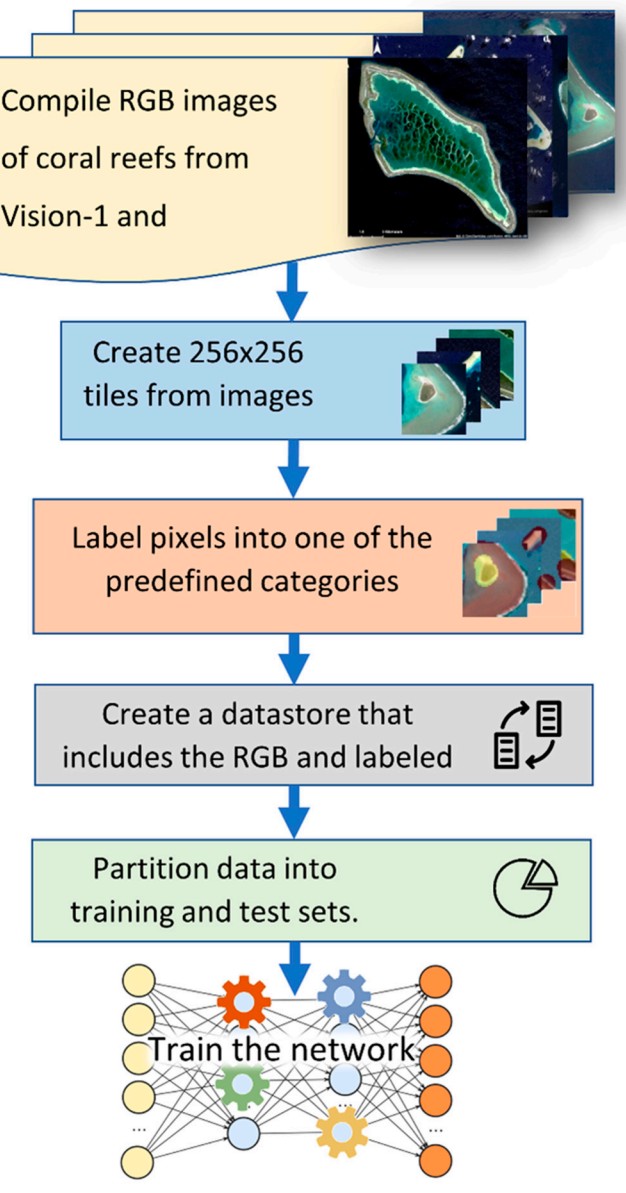

**Figure 3.** Flowchart showing the general steps followed for compiling and preparing data to train deep-learning network to identify black reefs.

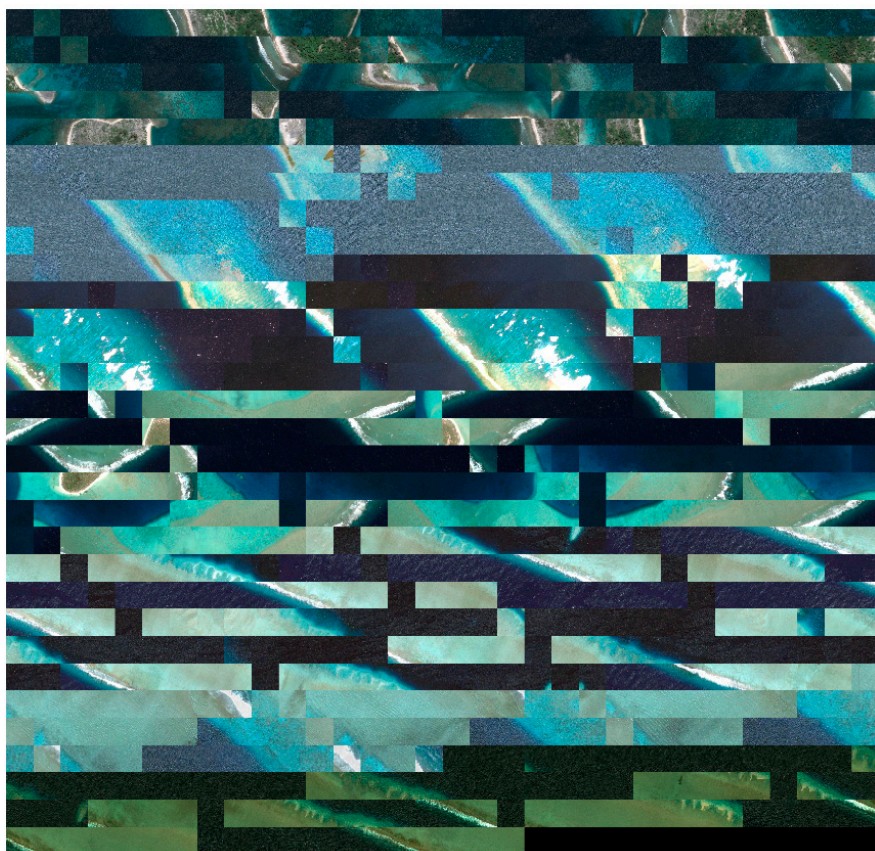

**Figure 4.** Using the ImageLabeler from MATLAB, 1600 images (256 × 256pixels) were labelled. The images consist of 4 Vision-1 images and 7 Google Earth images from 5 reefs (Kanton, Kenn, Kingman, Nikumamoro and Rose).

The data were labelled using the ImageLabeler program in Matlab 9.6 in three different classes: 'Black Reef', 'Non-Black Reef' and 'Water'. 'Black Reef' was the label for all known locations of shipwrecks that have created discoloration of reef regardless of shape and colour. 'Non-Black Reef' was used for all the other non-discoloured areas of the reef either on the surface or the submerged part of the reef. Finally, we labelled all areas which visually contained water but no reef as 'Water'. The labelled images include 3 channels (RGB) from different types of sensors with different resolution characteristics. The images were extracted from Google Earth and from Vision-1 as tiff files and stored in the form of an 8-bit monochromatic copy of the image, with 3 distinct intensity values reserved to represent each one of the 3 different classes, in addition to the null-intensity that represents the unlabelled regions of the image. Labelled pixels in each dataset corresponded to the classes of 'Black_Reef' by 0.071%; 'Non_Black_Reef' by 18.17%; and 'water' by 81.3%, (Figure 5). Due to the imbalance between the different classes, we weighed each class by the inverse of its frequency during the last layer of the classification networks (Pixel Classification Layer) to avoid bias in the learning process.

For training, the RMSProp (root-mean-square propagation) optimizer [71] and the Adaptive Moment Estimation (Adam) were tested. After multiple tests over both optimizers, we concluded that the RMSProp performed better, and the final quasi-optimal set of parameters we used to train the network are shown in Table 3.

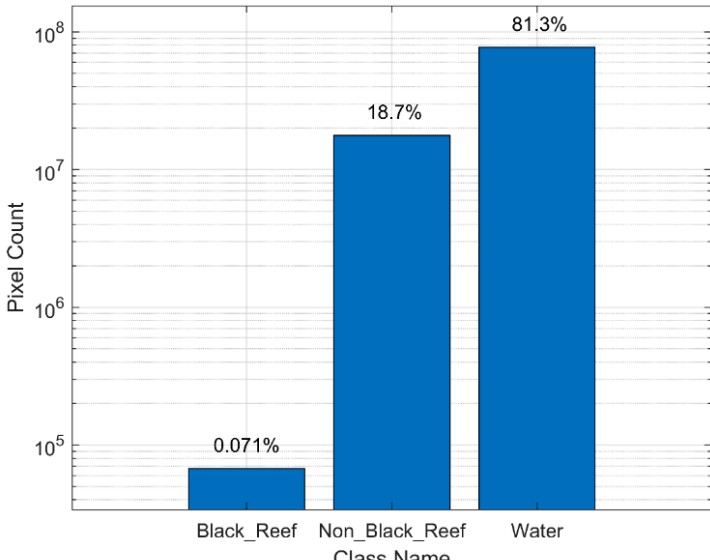

**Figure 5.** Histogram illustrating the number of pixels used in each of the 3 classes, 'Black_Reef', 'Non_Black_Reef' and 'Water'.

**Table 3.** Optimal set of parameters for the 8-layer networks.

| Parameter Name | Value |
| --- | --- |
| Squared Gradient decay factor | 0.9900 |
| Epsilon | $1 \times 10^{-8}$ |
| Initial learning rate | $3.0000 \times 10^{-4}$ |
| Gradient Threshold Method | l2 norm |
| Drop period | 5 |
| L2-regularization parameter | $1.0000 \times 10^{-4}$ |
| Gradient threshold | Inf (positive scalar) |
| Max epochs | 50 |
| Mini batch size | 8 |
| Shuffle | once |

For running the algorithm to train the data, we used the IRIDIS supercomputer at the University of Southampton. The training ran in parallel using 12 CPUs and one Node with 264 GB of main memory and took approximately 2 h to complete. For the calculations we used MATLAB 9.6 (2019b) software.

## 5. Results

After the learning procedure the trained network can be used to analyse and automatically segment new images of similar properties with the ones used in the training set and with a size of at least $256 \times 256$, which is the minimum used in the network (Figure 6). The segmentation via the trained model does not require significant computational resources. For example, a typical personal computer takes a few seconds for images of size $1600 \times 980$ pixels.

The performance of the trained network was evaluated for known black reef locations established by previous studies [18,24,58]. Due to the limited number of such known locations in the literature, and as at least 11 such black reefs occur in Kenn reef, we reserved certain regions that were not used in the training of the network exclusively for evaluation purposes. Additionally, we utilized the only known black reef occurrence in Rose reef for verification rather than training (Table 2; Figure 2). The trained algorithm was applied to four reefs, the Kenn reef, the Rose reef, the Kwajalein atoll, and the St Brandon reef. For the last two reefs, the presence of a shipwreck is known, but no black reef has been reported.

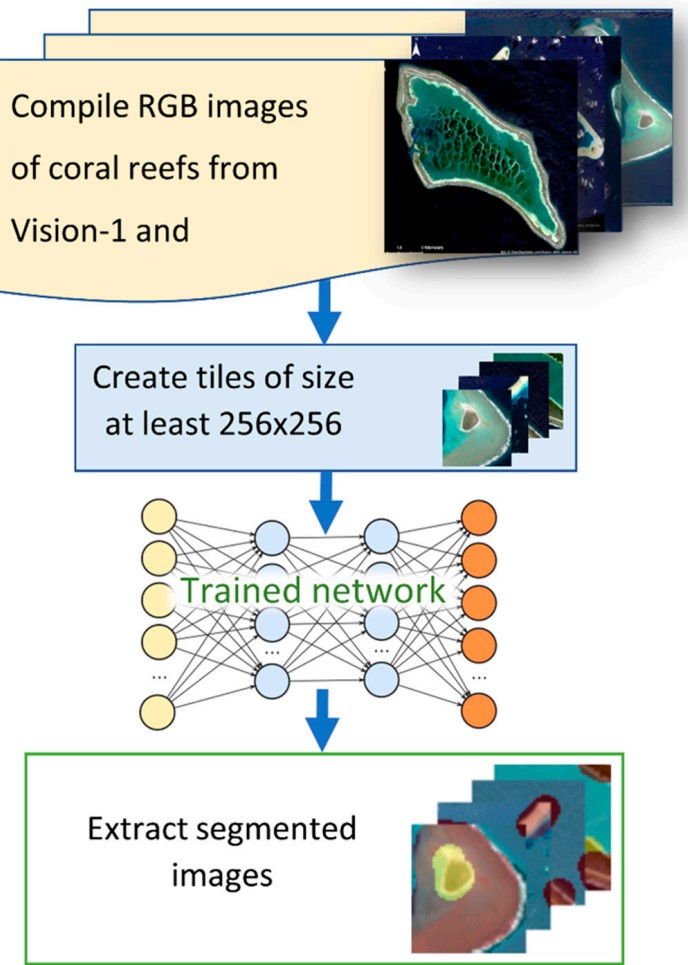

**Figure 6.** Flowchart showing the procedure of using the trained deep learning network to segment images.

*5.1. Kenn Reef*

In the case of Kenn reef, segmentation was applied to the whole image, consisting of both regions that were included and not included in training; however, evaluation focused on the seven black reef locations (Figure 7, locations 5 to 11) that were not used in training.

The resulting segmented image shows that, overall, water was clearly distinguished from land, including regions where clear, shallow waters allow visibility to the seabed. The algorithm managed to correctly identify the four known locations of the shipwrecks that were used for training, as expected. It also succeeded in identifying all seven remaining wreck locations that were not included in the training set.

It should be noted that although all wreck locations are clearly identified, the extent of each black reef is not mapped to its full extent as visible in the source RGB image. Instead, the algorithm performs best at identifying areas in close proximity to the wrecks, suggesting it identifies the most prominent area of discoloration. It should be noted that the resulting image has identified as black reef an area several hundred meters to the north of wreck 4. There are no additional data available to confirm the presence of an additional wreck/black reef source at this location, thus it is possible that it is a false detection.

Additionally, the different behaviour at locations 1, 5 and 6 is likely related to their proximity to or partial coverage by water. Furthermore, the algorithm marks as black reef several pixels located in a narrow zone along the swash, with the eastern side more continuous and the western more sporadic. This might be because Kenn reef is currently the only known reef with a high concentration of shipwrecks that have developed this discoloration. It is thus possible that the environmental impact/development of black reef has become more widespread. Confirmation of this, however, would require further

ground truthing, but is redolent of observations made during recent archaeological survey work, where the extent of black reef is linked to the spread of iron-rich objects out from wrecks [72].

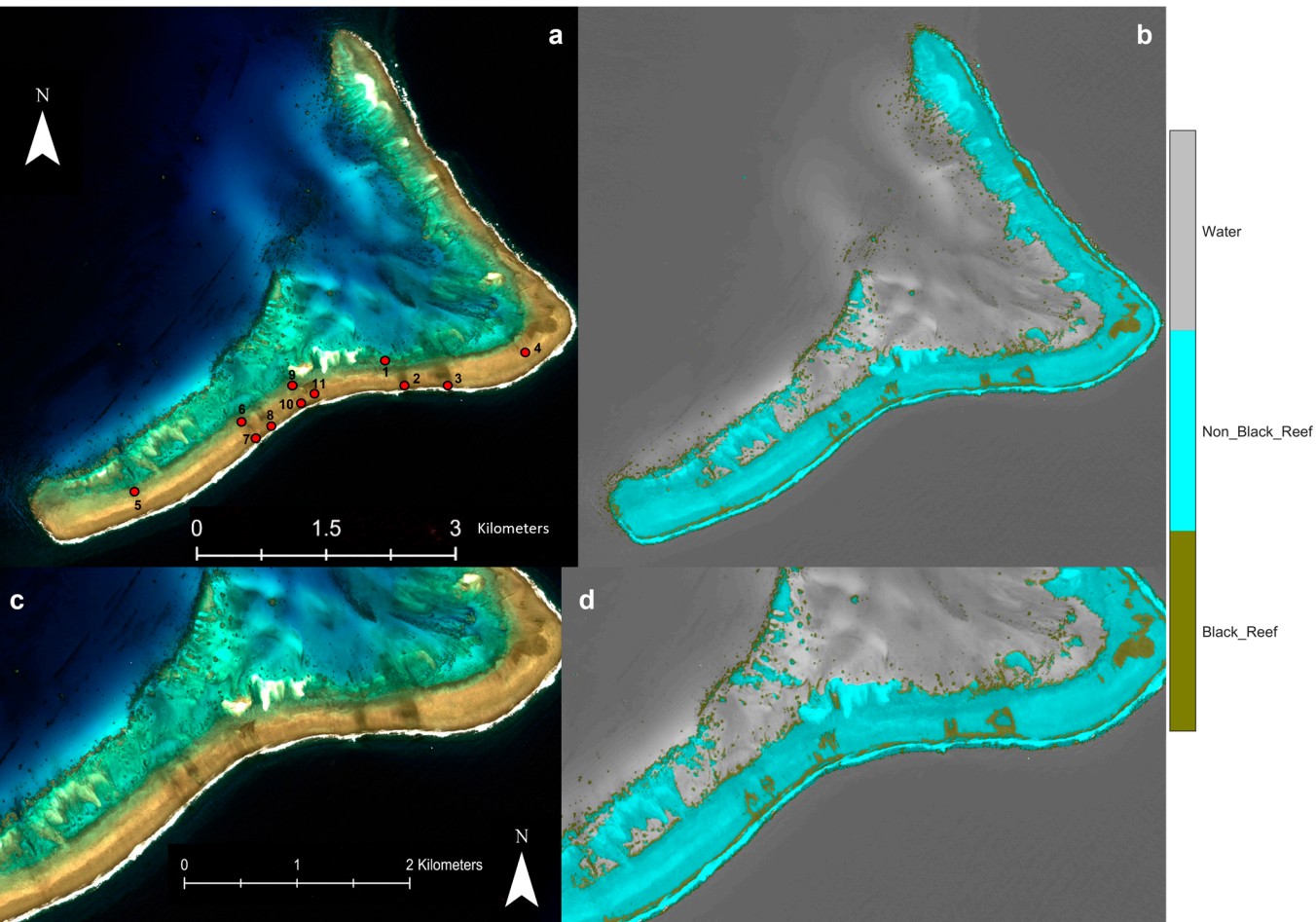

**Figure 7.** (**a**) Vision-1 satellite image of Kenn reef with the location of the 11 known shipwrecks (red dots), (**b**) the resulting segmented image after the application of the trained algorithm, (**c**) zoomed part of the Google Earth image and (**d**) the corresponding zoomed part of the segmented image.

### 5.2. Rose Reef

Rose reef has one known shipwreck. The Google Earth image includes a wide area that extends north-east from the location of the wreck that appears to be discoloured, suggesting the occurrence of a black reef. The part of the image used for training did not include this black reef area. Instead, a small part of the healthy reef was used for the training which does not include any known shipwreck or visible discoloration (Figure 8b).

The segmented image correctly identified several areas of discoloration, in the vicinity of the wreck. The pixels recognized as black reef extend north-east of the location of the wreck, reaching from the front reef crest area up to the back of the reef into the lagoon (Figure 8c,d).

The algorithm also classified as black reef several pixels at the upper left and right corners of the reef. This is clearly a false identification due to cloud shadows (Figure 9). Nevertheless, such false identifications can be readily avoided by masking the clouds and their shadows [73,74] prior to the segmentation.

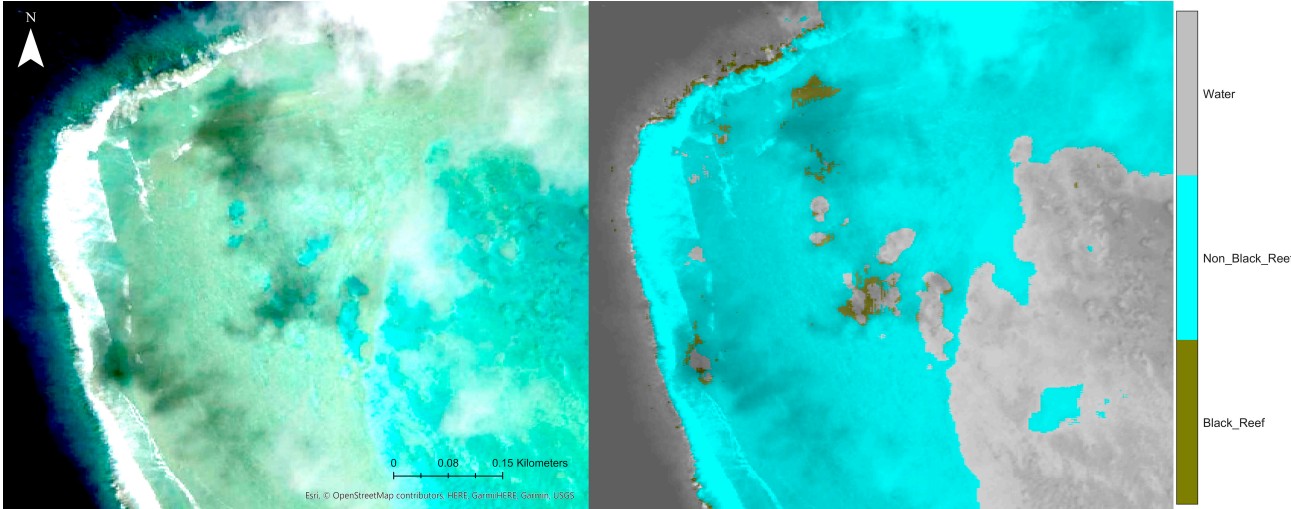

**Figure 8.** (**a**) Google Earth image (Google Earth Pro, 2011) from Rose reef with the location of the one known shipwreck (red dot), (**b**) the resulting segmented image after the application of the trained algorithm, (**c**) zoomed part of the location of the shipwreck showing the discoloration of the surrounding area, (**d**) the corresponding zoomed part of the segmented image.

**Figure 9.** (**Left**) a zoomed part of the Google Earth image (Google Earth Pro, 2011) of Rose reef (northwest corner) that shows the presence of clouds. (**Right**) the resulting segmented image where the shadows of the clouds have been falsely identified as black reefs.

### 5.3. Caroline Reef

Caroline reef has two shipwrecks; however, the exact location of only one of them is known and this is the one we used in this work.

We applied the trained algorithm to an extended part of the atoll where the shipwrecks are thought to be located. We used two Google Earth images from two different dates; one from 2004, which is the first clear image from the atoll, and the second one from 2011, which is the last image from this area. Sub-aerial vegetation was masked out by thresholding an NDVI calculated from a Landsat image (LC09_L1TP_055067_2021). The final segmented image from the year 2004 (Figure 10b) shows that around the shipwreck there is the appearance of a black reef reaching to southern side of the lagoon's entrance. The segmented image from 2011 (Figure 10d) shows that the black reef has spread to the other side of the entrance and extended further south along the reef front.

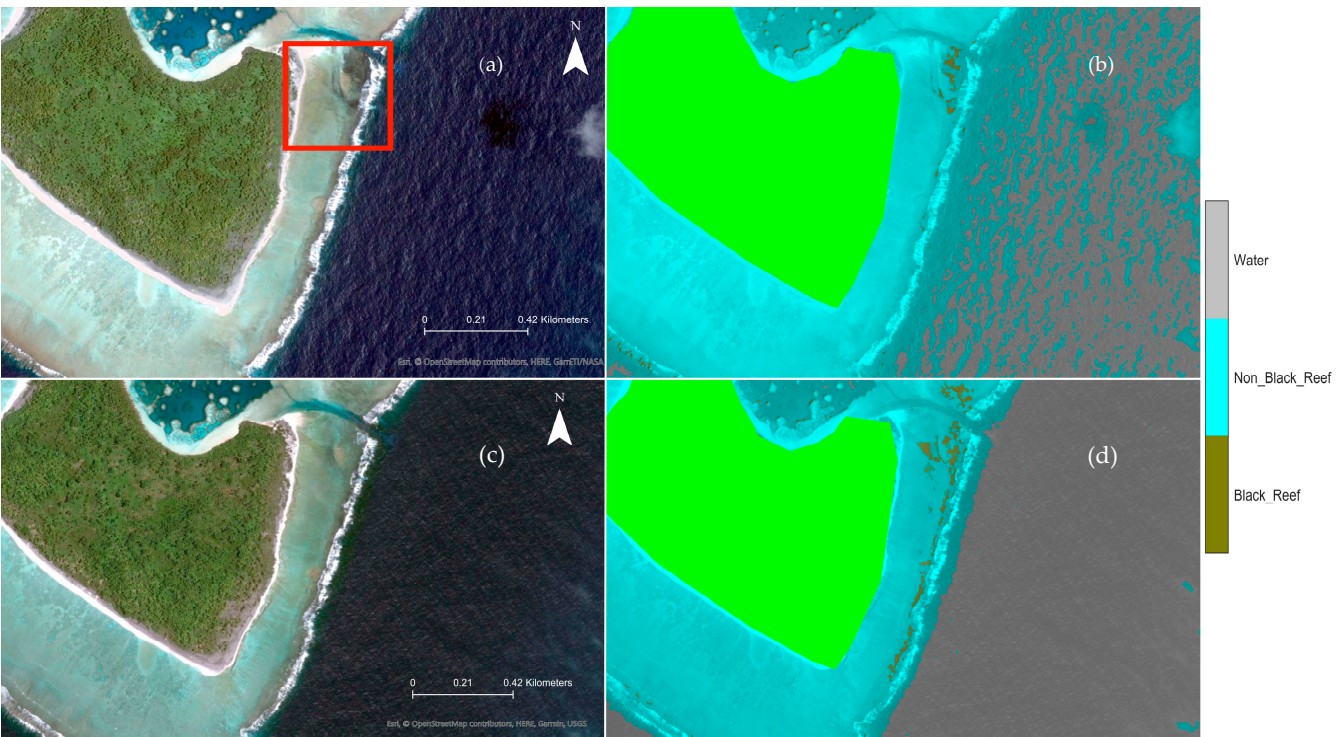

**Figure 10.** (**a**) Google Earth imagery from year 2004 (Google Earth Pro, 2004, 2011) and (**b**) the segmented image. The light green colour indicates subaerial vegetation mask based on Landsat imagery, while the red rectangle indicated the area where the shipwreck is located [18]. The segmented image shows the appearance of a black reef near the entrance of the lagoon. (**c**) Google Earth image taken in 2011 while in (**d**) is the final segmented image. From the segmented image is clear that the black reef has extended to the other site of the lagoon's entrance as well as across the coastal area.

### 5.4. Kwajalein Reef

At the Kwajalein reef we explored the only shipwreck that lies on shore at the northmost part of the island.

There are several high-resolution images available through Google Earth for this atoll (Figure 11, left column). The earliest image that depicts the shipwreck is from 2005 and shows the largely intact hull of the stranded vessel. Later images that are available from 2013, 2016, 2019 show the integrity of the vessel gradually deteriorating, and that parts of the ship's hull/deck/bridge have shattered. The most recent image is from 2022 and shows that the body of the vessel has largely broken into smaller pieces, some of which have visibly spread up to a few hundred meters further inshore. Additionally, for the first time there is visible a wide region of discoloration that resembles the occurrence of a black

reef, extending inshore, south-southwest with respect to the wreck. At present there is no literature referring to the presence of black reef at this location.

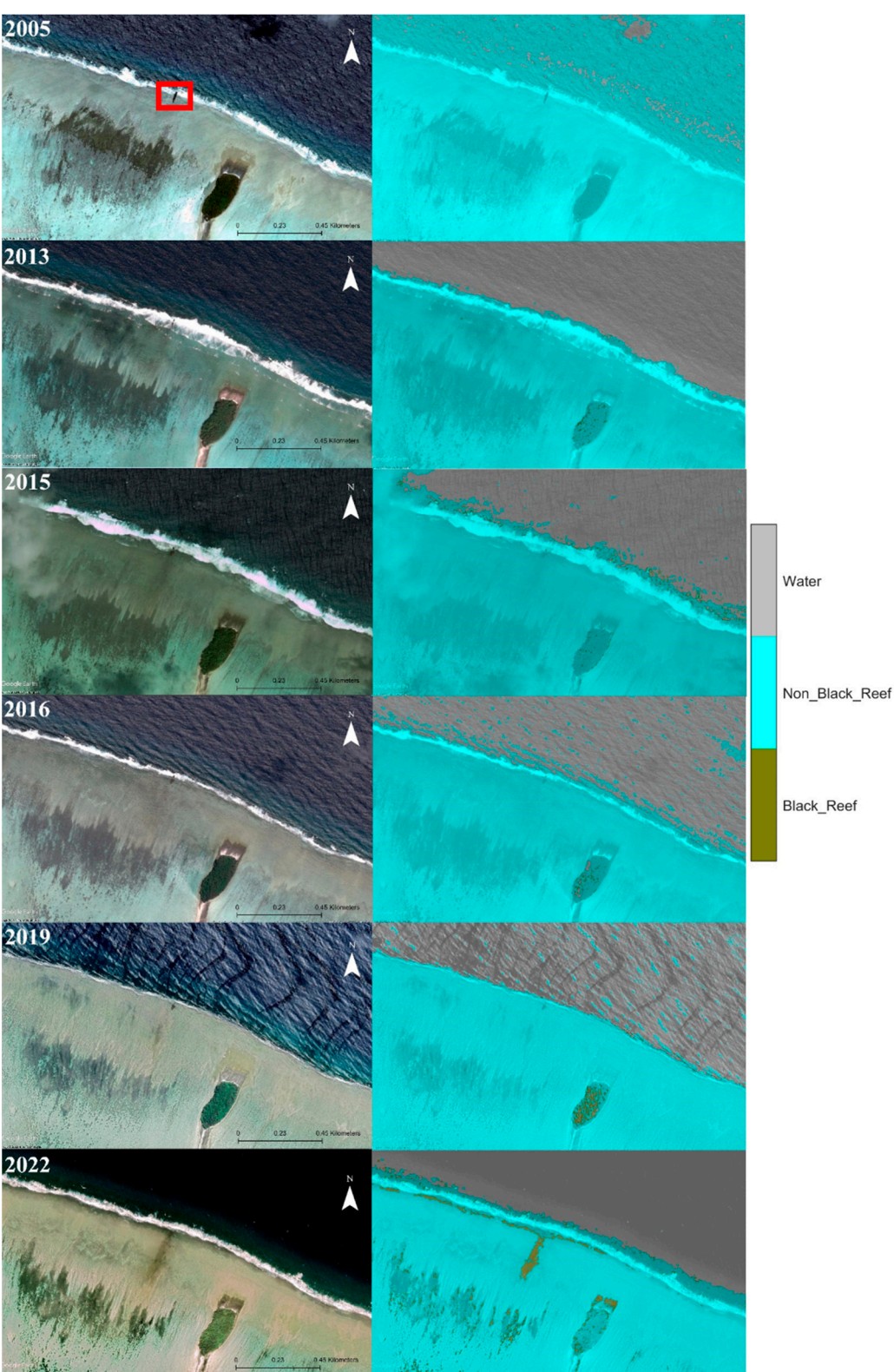

**Figure 11.** (**Left**) column: All available Google Earth images from the Kwajalein reef for the years 2005, 2013, 2015, 2016, 2019 & 2022, inside the red rectangle is the location of the shipwreck. (**Right**) column: the corresponding segmented images.

We applied the trained segmentation algorithm to all the images available from Google Earth (Figure 11, right column). For the images from 2013–2019, the algorithm does not show any occurrence of black reef near the shipwreck. In the 2022 image, however, the dark region is largely classified as a black reef. As with the other cases, a narrow region parallel to the shore that initiates from the shipwreck is also identified as a black reef. In this case, it appears that the westward expansion of this feature reaches substantially longer distances than its eastward counterpart, possibly related with the dominant currents in the region (Figure 12). Sporadic pixels classified as black reefs are also visible in vegetated areas. Such regions can be readily masked using the NDVI of multispectral images and be excluded from the interpretation.

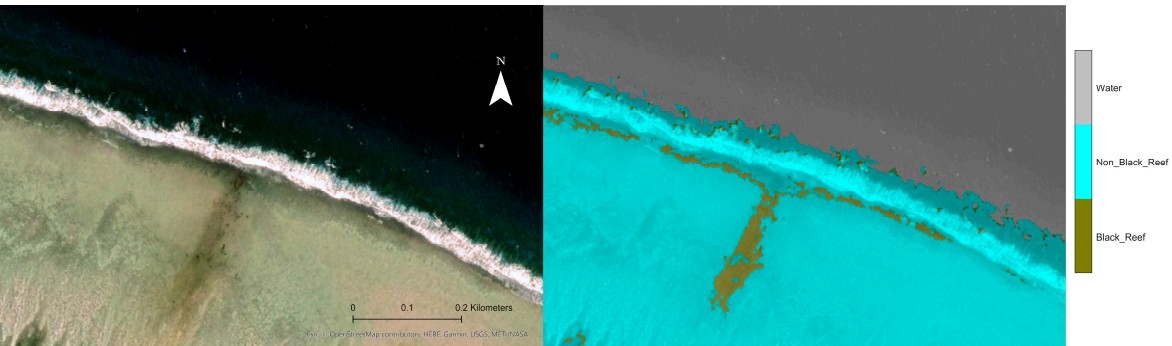

**Figure 12.** (**Left**) zoomed Google Earth imagery (Google Earth Pro, 2022) of the Kwajalein reef. (**Right**) corresponding segmented image.

*5.5. St Brandon Reef*

For the St Brandon reef, we explored the single wreck which has a known location.

We applied segmentation using the trained network on this image to check if there is any discoloration due to a possible development of algal bloom in the surrounding area. Figure 13a shows the Google Earth image and the shipwreck (inside the red rectangle) while Figure 13b is the segmented image. The segmentation has not identified any black-reef pixels around the wreck. Inspection of the image does not suggest any occurrence of discoloration and to the knowledge of the authors, no black reef has been reported for this location.

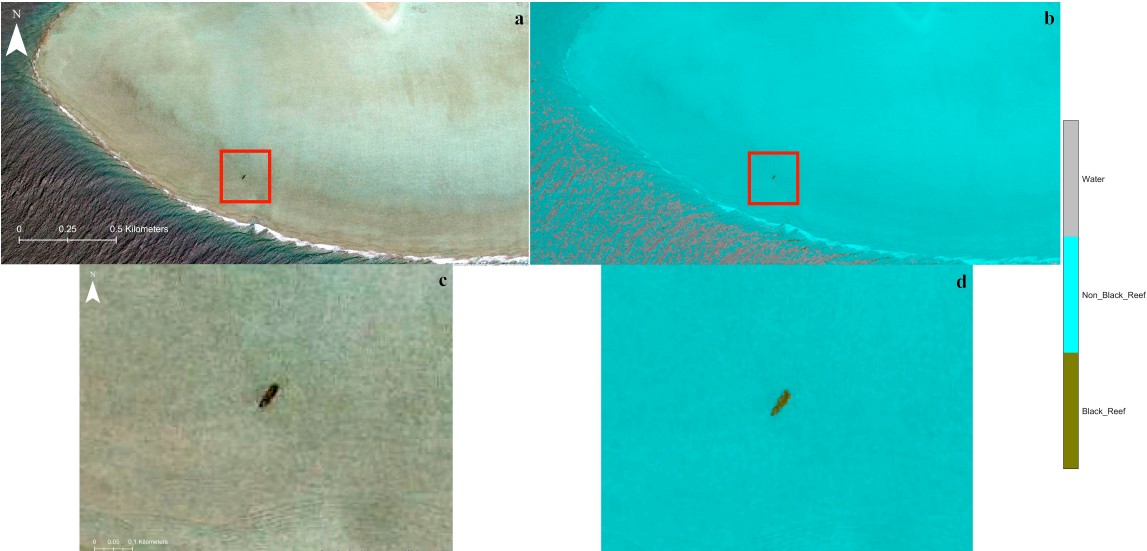

**Figure 13.** (**a**) Google Earth image of the St Brandon reef (Google Earth Pro, 2005); the red rectangle shows the location of a shipwreck, (**b**) segmented image, (**c**) zoomed part of the Google Earth image focusing on the shipwreck, (**d**) the corresponding zoomed part of the segmented image.

## 6. Discussion

The results obtained in this study are promising. First, they demonstrate that the automated detection of black reefs, and from this the presence of metal rich wrecks, in coral reef areas is possible, with two new potential black reefs identified (one on Kenn reef and one at Kwajalein reef). Second, it indicates how time series data can be used to monitor and measure impact. This is important as one of the challenges of survey in coastal and marine areas is the rate at which change occurs and the degree to which signatures may be obscured at one point in time but revealed at another [75]. The increasing availability of time series data offers a route to address this problem, but the data volumes involved mean that an element of automation will be required to make the task manageable.

In the case of Kenn reef the potential for identifying growing and cumulative impact is made clear, as is the need for time series data. The linear extent of black reef identified through image segmentation, matched to the large number of wrecks present on the reef suggests a cumulative and/or widening impact leading to a wider phase shift in environment. This stands in contrast to the more localized black reef 'halos' witnessed at other locations on Kenn reef and at the other reef sites examined in this project [23,27]. This observation is supported by comparing accounts from the two published archaeological surveys at the reef [22,72]. An initial survey in 1997 recorded an area of black reef approximately 50 metres by 25 metres [22], whereas a survey in 2017 indicated a larger black reef area of 157 by 60 metres [72]. They go on to note that accurately measuring the widening area of impact at this site would require further targeted fieldwork. From their published report and associated blog, the driver for the widening area seems likely to be the presence/dispersal of large, iron-rich objects away from the wreck.

Taken together the publication of these two surveys, which happened c. 30 years apart, demonstrate the role that remote identification and monitoring can have. The data presented here only permit a hypothesis, but when combined with insights from ground-based surveys, they allow confidence to grow. These combined results can be used by researchers, government agencies and charities when considering locations for further study or evaluation. The outputs from such work would in turn feed into improving the accuracy of automated detection and monitoring. Thus, even if the linear extent at Kenn Reef was found to be a false positive, additional ground truthing would serve to calibrate and refine future automated outputs. The same is true for the newly identified potential black reef sites; they offer an automated approach for targeting field-based research. Critically, the shortcomings of remote sensing with regards to certainty and the episodic nature of ground-based surveys highlight the need for both activities to occur, but most profitably hand-in-hand. This study would have been severely limited without the results of [22,72].

The results from Kwajalein reef demonstrate that severe deterioration of a shipwreck acts as a catalyst for the development of a black reef. For several years the existence of a shipwreck may not cause a significant growth; however, when a tipping point of break up is reached, black reef growth accelerates. This is supported by the field observations of [72], where the distribution of metal-rich artefacts and materials were seen to extend the reach of black reef.

While the results gained in this work are positive, it is also clear that there are a number of issues that need to be highlighted and addressed. First is the false identification of black reefs caused by shadows cast by clouds. This necessitates the masking of cloud cover areas prior to the application of segmentation, a time-consuming or data-limiting step. Second, as in many remote sensing tasks, identification of black reefs benefits from images with higher spectral resolution. The limited spatial resolution of Sentinel-2 and Landsat 7/8/9 satellites restricted identification to areas where the black reef extent ran into tens of meters. Thus, while the wreck itself does not need to be visible, limitations are still imposed by the spatial extent of the phenomenon to be observed. Thus, for early detection of black reef formation, or recognition of subtle changes through time, higher spatial and temporal resolutions are required. The implication of this is that while we have demonstrated that

data that are freely available today can be of use in the detection of black reefs, in many cases higher resolution (and thus commercial) sources are preferable.

Finally, in situ investigations, in combination with historical and archaeological research regarding the type of wrecks, their exact age, construction materials and their cargo are necessary to fully assess the archaeological signature of different vessels. Furthermore, this will shed light on how they are expected to interact with the environmental conditions and better understand the mechanism, propagation speed and the time scales of the contamination. The current indication is that the contamination can persist even after years of the wreck's removal [17,25] or be irreversible [18]. This suggests that even older wrecks could develop this discoloration in their surrounding environment if they contained some type of iron in their construction materials.

## 7. Conclusions

The detection of shipwrecks is an important task as, in addition to their archaeological significance, they can pose a substantial risk to the surrounding marine environment. Leakage of toxic or hazardous materials they may contain can be catastrophic for marine ecosystems, human life, local economies, and societies. In coral reefs, even in cases where no such materials are present, contamination with high amounts of iron from the building materials of the wrecked vessels can pose a significant threat for the reef itself through the formation of black reefs.

This work showed that detection of black reefs is possible using high-spatial resolution imagery, and deep learning models. More specifically, a simple eight-layer, fully convolutional network, trained with a limited number of labelled data from the known locations of black reefs was able identify potential locations of black reefs.

Moreover, through the detection of black reefs, the position of shipwrecks that are not currently visible can be inferred indirectly, as well.

By examining time series data, it is possible to monitor the expansion of the black reef and therefore the progress of the contamination of the reef with iron. As a wreck breaks down, the area impacted will grow, meaning that early detection and monitoring has a role to play in preventing ecological impacts.

Our results indicate that wind, ocean currents and wave action can transfer debris from the wreck to adjacent regions of the reef. Furthermore, it showed that the initiation of the black reef may be delayed for years in cases where the grounded ship's hull structural integrity is intact. Change will accelerate when the wreck starts deteriorating and breaks into pieces. Therefore, the detection of an expanding black reef can serve as a proxy for recognizing wrecks where a change of state is occurring.

As a community, our target should be to continue expanding the library of labelled data and to openly publish results from field-based research. As more locations of black reefs become available from different locations worldwide, the ability to confidently detect and monitor will grow.

**Supplementary Materials:** The following supporting information can be downloaded at: https://www.mdpi.com/article/10.3390/rs15082030/s1, Table S1: List of all known shipwrecks that have developed a discoloration in their surrounding environment developing a "black reef". Figure S1: The left image shows the Kenn reef (Vision-1) and right image is the zoomed part included in the red rectangle of the left image. The red dots indicate the location of the shipwrecks; Figure S2: The left image shows the Kingman reef (Google Earth imagery, 2013) and the right image is the zoomed part included in the red rectangle of the left image. The red dot indicates the location of the shipwreck [76,77].

**Author Contributions:** Conceptualization, A.K. and F.S.; methodology, A.K., F.S. and P.B.; software, A.K. and P.B.; validation, A.K., F.S. and P.B.; formal analysis, A.K., F.S. and P.B.; investigation, A.K.; resources, A.K. and F.S.; data curation, A.K.; writing—original draft preparation, A.K., F.S. and P.B.; writing—review and editing, F.S.; visualization, A.K.; supervision, F.S.; project administration, F.S.; funding acquisition, F.S. All authors have read and agreed to the published version of the manuscript.

**Funding:** This work was supported by the National Environment Research Council and the Daphne Jackson Trust.

**Data Availability Statement:** In this work we used openly available imagery provided from Jisc (https://www.jisc.ac.uk, accessed on 6 September 2020) and Google Earth (http://www.google.com/earth/index.html, accessed on 6 September 2020).

**Acknowledgments:** The authors would like to thank the Jisc (https://www.jisc.ac.uk, accessed on 6 September 2020) for providing us with Vision-1 imagery.

**Conflicts of Interest:** The authors declare no conflict of interest.

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
