# Peer review of "Identification of Black Reef Shipwreck Sites Using AI and Satellite Multispectral Imagery"

_remotesensing, doi:10.3390/rs15082030_

Round 1
Reviewer 1 Report
The topic of the paper is interesting and certainly worth doing from both environmental and archaeological standpoints. Wrecks can be both cultural heritage and sources of pollution, and from both heritage and environmental management there is a clear need to a) identify wreck locations and b) monitor wreck degradation over time. Any work pertaining to this and which can take advantage of the increasing availability of satellite imagery is therefore worth doing. I am not an expert in AI-based approaches, therefore I cannot comment on the underlying technicalities of the method. From a non-specialist / applied remote sensing perspective, it looks reasonable. However, I hope that one of the other reviewers will be able to confirm whether this indeed is the case.
Nevertheless, although I found the manuscript interesting and the topic worthy of study, the manuscript has a number of shortcomings which, in my opinion, prevent it from publication in its present form. As such I have recommended major corrections. My main points are summarized below with more detailed comments and suggestions added to the attached pdf.
1) Structure: the paper is hard to follow, particularly in terms of differentiating training sites and evaluation sites. The rationale for choice of sites is also not clearly expressed. For instance, the reader is given the impression from the start that the focus is on the southwest Pacific, and then suddenly an evaluation site from the Indian Ocean is thrown in. Elements of the method can also be found spread in different places. A simple (and conventional) structure along the lines of Methodological Background - Study Area/Sites – Materials and Methods – Results would be more appropriate.
2) Discussion: in its current state, the discussion is not well developed. It summarizes the results (which is really more something for the conclusion) and raises a couple of points, but does not expound on them in any detail. As a result, the manuscript feels like a report rather than a full scientific paper. For instance, the following points are all worthy of consideration:
· - The linear extent of black reef along the reef front (e.g. Kenn Reef). Compare and contrast this with the other study sites. Does this strengthen or refute the idea that this is genuine black reef or a false detection? And therefore does this make a case for more research?
· - Role of this technique in relation to detection of older wrecks (i.e. those considered to be of heritage/archaeological value). How long does the black reef phenomena persist? Will a reef recover and remove the signature? This requires a better consideration of wreck age and type in relation to the black reef – see comment below. This is potentially quite an important point because one of the key justifications in your paper is detection of wreck sites for heritage/archaeological purposes. Therefore if only the signatures of very modern wrecks are showing, then this reduces the relevance of the technique for heritage/archaeology.
· - Role of this technique in monitoring. Your Kwajelein and Caroline examples are potentially quite good examples of change over time. So expand on this – discuss further the potential expansion and usefulness of this technique given a) the remoteness of the areas and b) their ecological sensitivity.
· - Expansion to other datasets. You touch on Landsat and Sentinel but what about Planetscope? Can be accessed via research licences and has 3m resolution, and VERY high temporal resolution.
· - Black reef formation processes and links to wreck types and degradation. This point is raised in the conclusion but not discussed in detail in the discussion. This would require better information on the wrecks (see point below).
Even if you cannot come to a conclusive answer (e.g. can we detect older ‘heritage’ wrecks based on black reef,), at least show that you are thinking about the problem and suggesting ways to tackle it.
3) Wrecks: the treatment of wrecks is very superficial. Each wreck has only been considered in terms of being an object which could create a black reef. However, as the authors themselves point out, other factors (e.g. degradation) control this also. I can also think of other wreck-related factors which would also control this – e.g. presence/absence/type of cargo; construction/quantity of iron; date of loss/recency of wreck; age of vessel at time of sinking etc… I don’t mean that the authors need to compile a full wreck database, but there should be more of this information present in order to interpret, or suggest reasons for, the patterns they see. For example; the St. Brandon reef wreck – why might there be no black reef here? Could it be a lack of degradation? Or even the hull type? Obviously, this is difficult to do when you don’t know what the wreck is. Therefore, I wonder if the authors should focus more on evaluation examples where information about the wreck is known Therefore, they can at least start to assess if there are any links between wreck type, age etc.. and the spread of black reef. A good start would be to use the UKHO database or information on wrecksite.eu. In fact, I’m actually surprised wrecksite.eu has not been used for supporting information. I’m pretty sure someone at Southampton has access to/a subscription to it.
4) Presentation: elements of the figures, tables and captions can be edited to aid comprehension and understanding.
5) Language: The English language starts off OK (e.g., good abstract), then gets progressively worse in terms of typos, sentence structure and phrasing. I think there is an probably a non-native speaker writer issue here. However, I think one of the authors is a native speaker so they should be prepared to give the manuscript a final and thorough read to weed out these problems.
Overall, I think there is a reasonable body of underlying work. But it needs to be much more clearly set out and the implications, significance and questions raised by this work need to be more fully developed.

Author Response
Responses to Reviewers
We thank the associate editor and the reviewer for the constructive comments and suggestions. Below is our analytical response to the remarks.
Reviewer: 1
We have also responded in detail, directly on the pdf with the comments of the reviewer (remotesensing-2227208-review.pdf-included in the revised submission). Please, see attachments.
Comment
- Structure: the paper is hard to follow, particularly in terms of differentiating training sites and evaluation sites. The rationale for choice of sites is also not clearly expressed. For instance, the reader is given the impression from the start that the focus is on the southwest Pacific, and then suddenly an evaluation site from the Indian Ocean is thrown in. Elements of the method can also be found spread in different places. A simple (and conventional) structure along the lines of Methodological Background - Study Area/Sites – Materials and Methods – Results would be more appropriate.
Reply
We have updated the structure of the manuscript, changed the Figures and added an additional Table.
Comment
- Discussion: in its current state, the discussion is not well developed. It summarizes the results (which is really more something for the conclusion) and raises a couple of points, but does not expound on them in any detail. As a result, the manuscript feels like a report rather than a full scientific paper. For instance, the following points are all worthy of consideration:
- - The linear extent of black reef along the reef front (e.g. Kenn Reef). Compare and contrast this with the other study sites. Does this strengthen or refute the idea that this is genuine black reef or a false detection? And therefore does this make a case for more research?
- - Role of this technique in relation to detection of older wrecks (i.e. those considered to be of heritage/archaeological value). How long does the black reef phenomena persist? Will a reef recover and remove the signature? This requires a better consideration of wreck age and type in relation to the black reef – see comment below. This is potentially quite an important point because one of the key justifications in your paper is detection of wreck sites for heritage/archaeological purposes. Therefore if only the signatures of very modern wrecks are showing, then this reduces the relevance of the technique for heritage/archaeology.
- - Role of this technique in monitoring. Your Kwajelein and Caroline examples are potentially quite good examples of change over time. So expand on this – discuss further the potential expansion and usefulness of this technique given a) the remoteness of the areas and b) their ecological sensitivity.
- - Expansion to other datasets. You touch on Landsat and Sentinel but what about Planetscope? Can be accessed via research licences and has 3m resolution, and VERY high temporal resolution.
- - Black reef formation processes and links to wreck types and degradation. This point is raised in the conclusion but not discussed in detail in the discussion. This would require better information on the wrecks (see point below).
- Even if you cannot come to a conclusive answer (e.g. can we detect older ‘heritage’ wrecks based on black reef,), at least show that you are thinking about the problem and suggesting ways to tackle it
Reply
We have updated the discussion section and modified the text according to the suggestions of the reviewer.
Comment
3) Wrecks: the treatment of wrecks is very superficial. Each wreck has only been considered in terms of being an object which could create a black reef. However, as the authors themselves point out, other factors (e.g. degradation) control this also. I can also think of other wreck-related factors which would also control this – e.g. presence/absence/type of cargo; construction/quantity of iron; date of loss/recency of wreck; age of vessel at time of sinking etc… I don’t mean that the authors need to compile a full wreck database, but there should be more of this information present in order to interpret, or suggest reasons for, the patterns they see. For example; the St. Brandon reef wreck – why might there be no black reef here? Could it be a lack of degradation? Or even the hull type? Obviously, this is difficult to do when you don’t know what the wreck is. Therefore, I wonder if the authors should focus more on evaluation examples where information about the wreck is known Therefore, they can at least start to assess if there are any links between wreck type, age etc.. and the spread of black reef. A good start would be to use the UKHO database or information on wrecksite.eu. In fact, I’m actually surprised wrecksite.eu has not been used for supporting information. I’m pretty sure someone at Southampton has access to/a subscription to it.
Reply
We have included all available information to the knowledge of the authors. However, if the reviewer has any suggestions of any sources with additional information we are happy to included.
Comment
4) Presentation: elements of the figures, tables and captions can be edited to aid comprehension and understanding.
Reply
We have updated the Figure 1, Figure 2, Figure 8, we have modified Table 1 and added a Table 2. We have also included four additional Figures at the Supplementary Material, two flowcharts to explain the methodology followed in this project and two Figures with zoomed black reefs.
Comment
5) Language: The English language starts off OK (e.g., good abstract), then gets progressively worse in terms of typos, sentence structure and phrasing. I think there is an probably a non-native speaker writer issue here. However, I think one of the authors is a native speaker so they should be prepared to give the manuscript a final and thorough read to weed out these problems.
Reply
We have carefully reviewed the manuscript and corrected any language issues. We have also rephrased parts of the manuscript to make it clearer to the reader.

Reviewer 2 Report
A significant oversight of the paper is the verification of remote sensing data. Does it correspond to the reality on the ground? This requires further investigation. Perhaps this could be done in a follow-up study.
I advise the authors to check the consistency of the paper in terms of contractions, punctuation, and grammar usage.
153 should be Google Earth
518 edit caps.
Author Response
Responses to Reviewers
We thank the associate editor and the reviewer for the constructive comments and suggestions. Below is our analytical response to the remarks.
Reviewer: 2
Comment
I advise the authors to check the consistency of the paper in terms of contractions, punctuation, and grammar usage.
Reply
We have checked and corrected any contractions, punctuations, and grammar usage that we could find in the paper. We have also rephrased parts of the manuscript to make it clearer to the reader.
Comment
153 should be Google Earth
Reply
Corrected
Comment
518 edit caps.
Reply
Corrected
Reviewer 3 Report
I appreciate the opportunity to review the article titled Identification of black reef shipwreck sites using AI methods and remotely sensed data. After careful reading, I'd say the article has number of flaws which need to be fixed. Having said that, I have recommendations that I believe the authors will find useful and that, if implemented, would make this research suitable for publication in the journal.
(1) Abstract needs to be more elaborate. Result of this study needs to be mentioned explicitly and please provide implication of this study at the end of this section.
( 2) Introduction - The Introduction section should have been improvised, a description of other studies that are related, and most crucially, the research gap that guided the development of this specific research, authors introduced this topic without proper background. I would recommend to establish research idea of the article more explicitly. Without scientific referencing and detailed analysis of those approaches, the conclusion and argument of the article do not sound appropriate. Some of the background information need to be rearrange throughout the introduction section. As per the keyword second paragraph should be related to black reef but I do not see any relevant information on black reef, if it is not pertinent, please remove or rearrange. Please provide in text citation in lines 38-45, need to provide exiting studies on the given topic with a comparison and how this study is suitable than other techniques. In doing so if required Please rearrange the keywords so that the background of the study sounds appropriate. Relevant citations are missing in lines 27 to 37.
( 3) Entire methodology section is poor and vague which make the research unclear in terms of methodological design. It is hard to understand the direction of this research. It is necessary to reorganize the hierarchy of Materials and Methods such, overview of the method, Data Collection, and Data Analysis etc. Please include a brief flowchart to help readers who are unfamiliar with this topic comprehend the study methodology. I do not see any flowchart for this study, please provide a brief but robust flowchart so that reader can be benefited from the study. Writing is not scientifically appropriate in this section. Please shorten this section and only put relevant information. Please do not use the term in this paper ( line 83) rather use in this study.
( 4) Result section is somewhat unclear, authors need to provide an overview what they are going to demonstrate in this section before, provide a brief overview at the beginning of this chapter. Shorten and concise this section.
(5 ) This article does not have a proper discussion section from where the reader can get comparative discussion on data and techniques being utilized for this study and their result and implication for future study, limitation should be included in this study. Please provide more citation for comparison sort of discussion.
(6) In the supplementary section please provide a figure showing all known shipwrecks that have developed a discoloration in their surrounding environment developing a “black reef”. enlarge few of the spots if possible within the map
Author Response
Responses to Reviewers
We thank the associate editor and the reviewer for the constructive comments and suggestions. Below is our analytical response to the remarks.
Reviewer: 3
Comment
- Abstract needs to be more elaborate. Result of this study needs to be mentioned explicitly and please provide implication of this study at the end of this section.
Reply
We have updated the abstract.
Comment
(2) Introduction - The Introduction section should have been improvised, a description of other studies that are related, and most crucially, the research gap that guided the development of this specific research, authors introduced this topic without proper background. I would recommend to establish research idea of the article more explicitly. Without scientific referencing and detailed analysis of those approaches, the conclusion and argument of the article do not sound appropriate. Some of the background information need to be rearrange throughout the introduction section. As per the keyword second paragraph should be related to black reef but I do not see any relevant information on black reef, if it is not pertinent, please remove or rearrange. Please provide in text citation in lines 38-45, need to provide exiting studies on the given topic with a comparison and how this study is suitable than other techniques. In doing so if required Please rearrange the keywords so that the background of the study sounds appropriate. Relevant citations are missing in lines 27 to 37.
Reply
We have updated the introduction, added additional information and relevant references, as suggested. The keywords have also been re-arranged.
Comment
3) Entire methodology section is poor and vague which make the research unclear in terms of methodological design. It is hard to understand the direction of this research. It is necessary to reorganize the hierarchy of Materials and Methods such, overview of the method, Data Collection, and Data Analysis etc. Please include a brief flowchart to help readers who are unfamiliar with this topic comprehend the study methodology. I do not see any flowchart for this study, please provide a brief but robust flowchart so that reader can be benefited from the study. Writing is not scientifically appropriate in this section. Please shorten this section and only put relevant information. Please do not use the term in this paper ( line 83) rather use in this study.
Reply
We have updated the methodology section and included two flowcharts in the Supplementary Material.
Comment
4) Result section is somewhat unclear, authors need to provide an overview what they are going to demonstrate in this section before, provide a brief overview at the beginning of this chapter. Shorten and concise this section.
Reply
We have updated the Results.
Comment
5) This article does not have a proper discussion section from where the reader can get comparative discussion on data and techniques being utilized for this study and their result and implication for future study, limitation should be included in this study. Please provide more citation for comparison sort of discussion.
Reply
We have updated the Discussion Section
Comment
6) In the supplementary section please provide a figure showing all known shipwrecks that have developed a discoloration in their surrounding environment developing a “black reef”. enlarge few of the spots if possible within the map.
Reply
We have updated the Supplementary Material. We have updated Figure 2 in the main manuscript to show the locations of the shipwrecks and consequently the back reefs. We have also included two zoomed images of two reefs that show the presence of black reefs. As it is stated in the paper, unfortunately, we don’t have available imagery for all known reefs (list of reefs in Supplementary Material) that have developed this discoloration or we don’t have the information of where exactly this is located on the reef. Therefore, we show (and used in the research) only the reefs for which such information is available.
Round 2
Reviewer 1 Report
Thanks for revising the manuscript; I appreciate the time and effort which the authors have taken to do so. I can confirm that the manuscript reads better and is easier to follow. I am also happy to see that the discussion has been re-written and now raises some good discursive points. However, even from a relatively fast read I still spotted a number of typos. I advise a VERY thorough final read through to weed these out. A couple of points of clarification and enhancement are also necessary (see attached pdf). Therefore, I am recommending minor revisions. Overall though, it has definitely improved from the first submission, it just needs a few tweaks before it is ready for publication.

Author Response
We thank the reviewer for the constructive comments and suggestions.
We have included our detailed responses, directly on the pdf with the comments of the reviewer.
Furthermore, we include the annotated manuscript with the corrections for the reviewer.

Reviewer 3 Report
I was not able to see the flowcharts which is in the supplimentary section, anyway i would recommend to put the flowcharts in the main manuscript and rearrange writeup accordingly. Otherwise i am happy with the corrections.
Author Response
We thank the reviewer for the constructive comments and suggestions.
We have moved the flowcharts from the supplementary material to the main text as suggested.
Furthermore, we include the annotated manuscript with the corrections for the reviewer.